# Enhancing Cooperative Problem-Solving in Sparse-Reward Systems via Co-evolutionary Curriculum Learning

## Abstract

Sparse reward environments consistently challenge reinforcement learning, as agents often need to finish tasks before receiving any feedback, leading to limited incentive signals. This issue becomes even more pronounced in multi-agent systems (MAS), where a single reward must be distributed among multiple agents over time, frequently resulting in suboptimal or inconsistent learning outcomes. To tackle this challenge, we introduce a novel approach called Collaborative Multi-dimensional Course Learning (CCL) for multi-agent cooperation scenarios. CCL features three key innovations: (1) It establishes an adaptive curriculum framework tailored for MAS, refining intermediate tasks to individual agents to ensure balanced strategy development. (2) A novel variant evolution algorithm creates more detailed intermediate tasks. (3) Co-evolution between agents and their environment is modeled to enhance training stability under sparse reward conditions. In evaluations across five tasks within multi-particle environments (MPE) and Hide and Seek (Hns), CCL demonstrated superior performance, surpassing existing benchmarks and excelling in sparse reward settings.

## 1 Introduction

Deep Reinforcement Learning (DRL) has shown promising results in addressing various challenges within Multi-Agent Systems (MAS), including applications in robotics (Abbass et al., 2021)perrusquia2021multi, gaming (Rashid et al., 2020), and autonomous driving (Shalev-Shwartz et al., 2016). However, in sparse reward environments, reinforcement learning methods still face limitations regarding incentivization and learning efficiency, as agents typically receive feedback only upon task completion. This feedback mechanism dramatically increases agents' difficulty in efficiently exploring their environment and extracting meaningful insights from their actions.

In order to deal with the challenge of sparse reward, several methods have been proposed to improve the exploration efficiency, including reward reshaping (Ng et al., 1999; Hu et al., 2020), learning from demonstrations (Ross et al., 2011), policy transfer (Duan et al., 2017), and curriculum learning (Florensa et al., 2017; 2018). The core goal of these strategies is to enhance the agent's exploration ability by reinforcing the reward signal during training. Although these methods have achieved remarkable results in single-agent environments, their application in MAS faces many challenges. In MAS, the interactive decision-making of multiple agents often enhances environment dynamics and the sharp expansion of state space, which often weakens the effectiveness of the above strategies.

However, it is worth noting that most of the above methods have shown satisfactory performance performance in single-agent tasks. However, once entering the environment of a Multi-Agent System (MAS), the coexistence of multiple decision-making entities inevitably leads to the reduction of environmental stability (Bloembergen et al., 2015; Buşoniu et al., 2010) and the rapid expansion of state space (Hernandez-Leal et al., 2019). Such changes often considerably weaken the original effectiveness of these strategies and may even make them invalid.

This paper proposes a new framework called Coevolving Multidirectional Curriculum Learning (CCL), which aims to deal with cooperative decision-making problems in sparse-reward MAS. CCL innovatively combines automatic Curriculum learning (ACL) technology, which automatically generates and prioritizes intermediate tasks to minimize the dependence on and bias from exist-

ing knowledge. Unlike traditional ACL methods, CCL is uniquely designed to optimize two key dimensions in MAS:

**1. Refining Reward Granularity to Enhance Agent Interaction:** CCL implements finer-grained control by focusing on the unique perspective of individual agents during task generation. By leveraging a variational evolutionary algorithm, CCL can precisely decompose complex intermediate tasks in MAS, enabling balanced and optimized strategy development for each agent while simultaneously working toward the overall objective.

2. Cooperative Iteration to Enhance Training Stability: CCL employs an evolutionary algorithm based on cooperative co-evolution (Antonio & Coello, 2017), allowing for the simultaneous advancement of intermediate task evolution and agent policy development. This synchronization ensures task difficulty aligns with the agent's current skill level, preventing inefficient training caused by either too complex or too simple tasks. As a result, this approach facilitates the co-evolution of tasks and agent capabilities, significantly improving training stability and efficiency.

In a comprehensive experimental evaluation, CCL exhibits outstanding performance, surpassing existing curriculum learning approaches and achieving industry-leading results across five cooperative multi-agent tasks with sparse rewards.Further ablation experiment analysis reveals that compared with the traditional evolutionary methods in the field of MAS, the variational individual-perspective evolutionary algorithm adopted by CCL shows significant advantages, which verifies its unique value in improving the cooperation efficiency and effect of multi-agent systems.

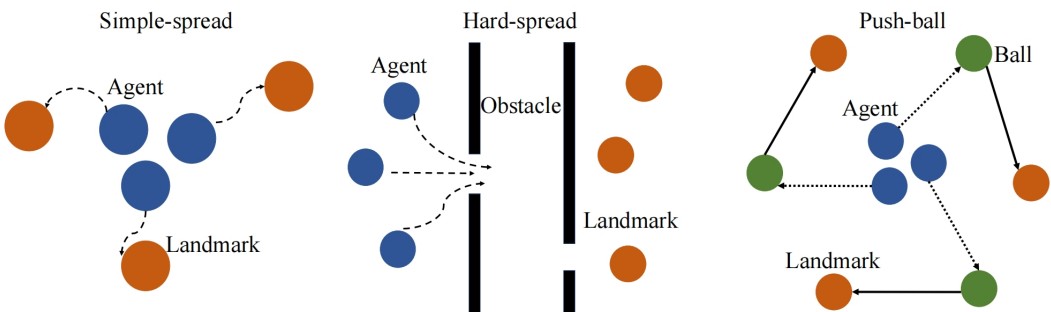

Figure 1: MPE is validated with three different collaborative task scenarios.

## 2 PROBLEM STATEMENT

In the reinforcement learning framework, the reward signal is a vital feedback mechanism that helps the agent assess its actions at each time step. By utilizing the Bellman equation (Kaelbling et al., 1996), the agent can develop a policy to maximize long-term cumulative rewards. As the reward function defines the core objective of the learning task and provides a quantitative measure of the agent's behavior, it is crucial to design a well-structured and compelling reward system. It is essential to recognize that agents may still follow suboptimal or inefficient action paths even with a clearly defined reward function. However, a well-designed reward function can significantly facilitate the agent's ability to learn and converge toward an optimal policy (Dewey, 2014).

However, designing an appropriate dense reward function faces many challenges in the complex MAS environment. This is mainly because the setting of dense rewards is often limited by researchers' prior knowledge, making it challenging to comprehensively cover all possible interaction scenarios and dynamic changes. In contrast, the sparse reward setting provides a more flexible and effective solution, which only provides a single reward signal when MAS achieves a predefined critical goal state $g$ (Booth et al., 2023), to get rid of the excessive dependence on the prior knowledge of researchers and effectively overcome the limitations of dense reward design.

In non-sparse reward settings, at each time step $t$, the agent observes its current state $s_t \in S$ and selects an action $a_t \in A$ based on its policy $\pi(a_t|s_t)$. The chosen action results in a transition to a new state $s_{t+1}$, determined by the environment's transition dynamics $p(s_{t+1}|s_t, a_t)$, and an associated reward $r_t$ is obtained from the reward function $r(s_t, a_t, s_{t+1})$. The sequence of states,

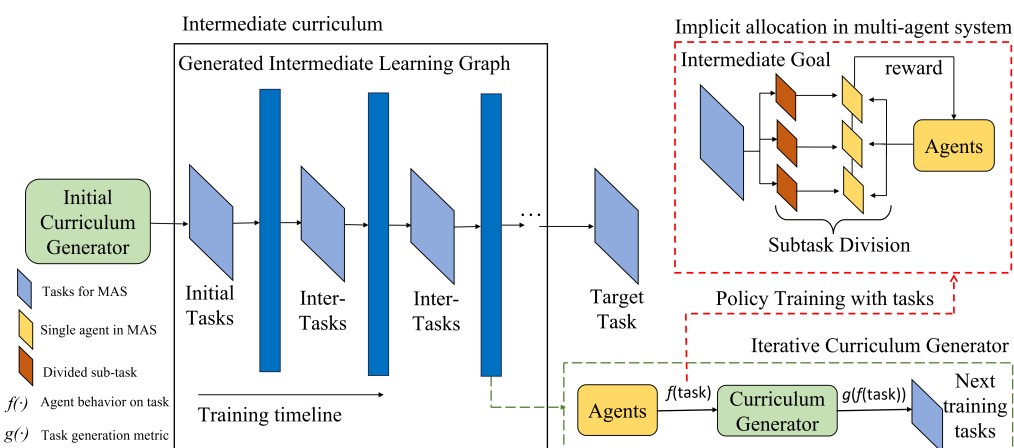

Figure 2: Compared with the course generation in the single-agent scenario, the intermediate task generation in the MAS is more complex because it contains sub-tasks with individual perspectives. In the sparse reward setting, the reward signal is shared among the agents, so it becomes particularly necessary to introduce a novel individual perspective mechanism to generate intermediate tasks.

actions, following states, and rewards over an episode of $T$ time steps form the trajectory $\tau = (s_t, a_t, s_{t+1}, r_t)_{t=0}^{T-1}$, where $T$ is either determined by the maximum episode length or specific task termination conditions. This outlines the process of reinforcement learning for a single agent.

The goal of this individual agent is to learn and maximize its expected cumulative rewarded policy:

$$J = \mathbb{E}_\pi \left[ \sum_t \gamma^t r_t \right] \tag{1}$$

where $\gamma$ is the discount factor, representing future rewards' diminishing value refinement degree of the optimization process is carried out by each time step inside the trajectory, that is, the optimization granularity is accurate to each time step.

However, the system dynamics significantly intensify when extending this general framework to MAS under sparse reward conditions. In this system, there are $N$ decision-making agents, where each agent $i$ takes an action $a_i$ at time step $t$ based on the observed state information and following its dedicated policy $\pi_i(a_i|s_i)$. The global state $s_t$ of the system is composed of the joint states of all individual agents, denoted as $s_t = (s_1, s_2, \ldots, s_n)$. Correspondingly, the joint action $a_t$ at each time step is also formed by the combination of actions from all agents, i.e., $a_t = (a_1 a_2, \ldots, a_n)$. In the sparse reward environment, reward signals only emerge when the system achieves specific predefined goal states, posing more significant challenges for agent collaboration and strategy optimization.

In cooperative multi-agent tasks, the goal of each agent is no longer focused on maximizing its reward but instead shifts toward optimizing the cumulative reward of the entire system. This requires agents to collaborate effectively, coordinating their actions to achieve the shared objective, thereby improving the overall performance of the multi-agent system. Consequently, the objective function $J$ for each agent $i$ is transformed into $J_i(\pi_i) = \mathbb{E}_{\pi_i} \left[ \sum_t \gamma^t r_i(s_t, a_t) \right]$, where $r_i(s_t, a_t)$ represents the reward received by agent $i$ at time step $t$ given the state $s_t$ and joint action $a_t$. The overall goal of the multi-agent system (MAS) then becomes the sum of the individual objectives, denoted as $J = \sum_i J_i(\pi_i)$.

At this point, it becomes evident that the essence of a multi-agent reinforcement learning algorithm lies in utilizing the rewards earned by all agents to optimize the overall collaborative strategy. However, this challenge is significantly heightened in a sparse reward environment, where agents receive limited feedback, making it difficult to effectively guide their actions and improve coordination toward the collective goal. In the case that there are only very few 0-1 reward signals, the total reward of the system can be simplified to a binary function:

$$r(s_t, g) = \begin{cases} 1, & s_t = g \\ 0, & \text{Otherwise} \end{cases} \tag{2}$$

With the increasing number of agents, the variance of MAS during training will increase exponentially. At the same time, in the sparse reward environment, each agent $i$ needs to achieve a sub-goal $g_i$ that is closely related to the overall goal $g$. This requirement often makes it difficult for the agent to get any reward feedback during training, thus increasing the difficulty of training. The following expression further illustrates the challenge MAS faces in obtaining effective feedback under sparse reward conditions:

As a result of the phenomenon above, agents face significant challenges during the exploration phase, leading to instability throughout the training process. This instability often causes the failure of many single-agent methods that typically perform well in sparse reward scenarios as they struggle to adapt to the multi-agent context where coordination and shared rewards are critical. To overcome this problem, this paper proposes CCL.

$$s_t = g \iff \forall i \in n, s_i = g_i \tag{3}$$

## 3 RELATED WORK

### 3.1 CURRICULUM LEARNING

To deal with the exploration problem caused by sparse rewards, the field of reinforcement learning has introduced a variety of strategies, including Reward Shaping (Laud, 2004), Intrinsic Motivation (Barto, 2013), and Curriculum Learning (Bengio et al., 2009). Among them, reward shaping and intrinsic motivation both supplement the original task reward information in the way of gain; that is, they increase the density of the reward in an additional way. Curriculum learning, on the other hand, takes a different divide-and-conquer strategy. It breaks the complex subject task into a series of more manageable subtasks to solve and sorts them according to certain logic to achieve the goal step by step. Given the high similarity between this step-by-step training mode and curriculum design in education (Abbass et al., 2021; Rohde & Plaut, 1999; Elman, 1993), the strategy is vividly termed Curriculum Learning.

Within the field of reinforcement learning, the core of the curriculum learning framework can be deconstructed into three pillar elements (Narvekar et al., 2017): first, the generation of tasks; second, the ranking strategy for these tasks; and finally, the application of transfer learning. These key components' construction process can be guided by the power of automated generators (Florensa et al., 2017) and expert domain knowledge. However, given the biases or limitations that may be implied in expert domain knowledge (Wang et al., 2019; Cobbe et al., 2019), the schedule Adaptive Automatic Course Learning (ACL) method shows superior performance in the face of complex and changing scenarios. The CCL method discussed in this paper is one in the frontier field of ACL.

The core challenges of ACL focus on the selection of metrics and the optimization of computational efficiency. Specifically, ACL aims to achieve immediate and accurate evaluation (Ren et al., 2018; Wu et al., 2024; Wang et al., 2024) of intermediate tasks without the assistance of expert knowledge. However, the current methods are generally rough in metrics, making it challenging to deal comprehensively with the complex policy hierarchy in MAS. In addition, whether it is a comprehensive evaluation of all intermediate tasks or the replay technology based on regret mechanism (Samvelyan et al., 2023; Parker-Holder et al., 2022), it may bring high computational cost. Therefore, in practical applications, ensuring the quality of tasks while reasonably controlling the computational cost has become an urgent problem that needs to be solved.

### 3.2 EVOLUTIONARY REINFORCEMENT LEARNING

Evolutionary Algorithm (EA) is a technique for solving optimization problems based on the natural evolutionary mechanism. Starting from a set of initial candidate solutions (i.e., population), they

retain the best individuals through the selection process based on the fitness evaluation of each solution and then use mutation and recombination operations to continuously generate new populations in the iterative process to approach the optimal solution (Beyer & Schwefel, 2002) gradually.

Given the excellent performance shown by reinforcement learning and evolutionary algorithms in their respective fields, the research on their integration has always been the focus of academic attention (Miconi et al., 2020; Pagliuca et al., 2020; Jianye et al., 2022). The introduction of an evolutionary algorithm aims to make up for the critical shortcomings faced in the process of reinforcement learning, such as the problem of long-term reward information (Samvelyan et al., 2023; Parker-Holder et al., 2022) backtracking and the lack of strategy diversity (Long et al., 2020). However, this integration path is not smooth, and a series of challenges accompany it. On the one hand, the high computational cost caused by the large population size (Wang et al., 2019) becomes a non-negligible obstacle. On the other hand, how to effectively retain and utilize the environmental information to prevent its loss in the evolutionary operation and encoding stage of evolutionary reinforcement learning is also an urgent problem to be solved.

## 4 METHODOLOGY

### 4.1 THE VARIATIONAL INDIVIDUAL-PERSPECTIVE EVOLUTIONARY OPERATOR

In this chapter, we will dive into each of the building blocks of CCL. The architecture of CCL is a coevolutionary system, and its core is composed of two closely related parts. The Agents are trained by the widely used Multi-Agent Proximal Policy Optimization (MAPPO) algorithm (Yu et al., 2022). Due to the popularity and maturity of the MAPPO algorithm, the specific details of the MAPPO algorithm are not explained here. For a comprehensive understanding of the working mechanism of CCL, please refer to Algorithm 1, which elaborates on the complete flow of CCL from startup to execution.

### 4.2 THE VARIATIONAL INDIVIDUAL-PERSPECTIVE EVOLUTIONARY OPERATOR

In this section, we comprehensively explain the critical components of CCL. As a co-evolutionary system with two primary components, the agent training process leverages the existing Multi-Agent Proximal Policy Optimization (MAPPO) algorithm Yu et al. (2022), which we will not cover in detail here. The entire workflow of the CCL algorithm is provided in Algorithm 1.

**Evolutionary Curriculum Initialization** At the start of training, a MAS (Multi-Agent System) often needs better initial policy performance, making it difficult for agents to complete complex tasks successfully. To address this, minimizing the task individual's norm in the initial population is crucial. Given the initial task domain $\Omega_0$, the randomly initialized task population should satisfy the following conditions. Here, $d$ denotes the Euclidean distance between the agent and the task, and $\delta$ is a robust hyperparameter, commonly set to around one percent of the total task space size.

$$\frac{1}{|\Omega_0|} \sum_{t_i \in \Omega_0} d(s_i, g_i) < \delta \tag{4}$$

**Task Fitness Definition** In previous methods, task evaluation primarily relied on the agent's performance in the environment Wang et al. (2019); Song & Schneider (2022); Parker-Holder et al. (2022), or utilized fundamental boolean values or intervals to filter tasks Racaniere et al. (2019). However, as we aim for agents to train on functions that present a balanced level of difficulty—neither too easy nor overly complex—these approaches often fail to capture the non-linear dynamics that affect task quality and success rates. Ideally, tasks with success rates close to 0 or 1 are deemed unsuitable for training. As success rates move from the midpoint to the extremes, task quality declines, quickly shifting from moderate difficulty to irrelevance. To better reflect this non-linear relationship, we introduce a sigmoid-based fitness function, which evaluates the appropriateness of tasks based on the agent's current performance, where $r$ represents the agents' average success rate on task $t$.

$$\tilde{f} = \frac{1}{1 + e^{-2|r-0.5|}} \tag{5}$$

**Variational Individual-perspective Crossover** In a MAS, the reward signal is distributed across multiple dimensions, especially when considering the perspectives of different agents, which can

---

**Algorithm 1** Coevolving Multidirectional Curriculum Learning

---

**Require:** training episodes $N$, curriculum population at episode $i$: $C_i$, total number of tasks $n_p$, task samples $n_t$, initial task region $\Omega_0$, soft selection rate $\alpha$, number of prototypes $k$
1: Initialize $C_0$ by uniformly sampling $n_p$ tasks from $\Omega_0$
2: Sample $n_t$ tasks from $C_1$
3: Initialize MAS policy $\theta$
4: **for** $i \leftarrow 1$ **to** $N$ **do**
5:     Train MAS on all $n_t$ tasks in parallel
6:     $r_j \leftarrow$ success rate of task $j$, where $j = 0$ to $n_t$
7:     **Remove underperforming tasks**
8:     $\tilde{f}_j \leftarrow \frac{1}{1+e^{-2|r_j - 0.5|}}$, for $j = 0$ to $n_t$
9:     $\tilde{f}_{\text{all}} \leftarrow$ **k-prototype fitness evaluation**$(f_1, f_2, \ldots, f_{n_t})$
10:     Create an empty curriculum population $C_{i+1}$
11:     **for** each task pair $c_k, c_{k+\frac{n_p}{2}}$ in $C_i$ **do**
12:         **if** random noise $\delta \sim (0, 1) > 0.5$ **then**
13:             $offspring \leftarrow$ **Multi-directional Cross**$(c_k, c_{k+\frac{n_p}{2}}, \tilde{f}_{\text{all}})$
14:             Add $offspring$ to $C_i$
15:         **else**
16:             $offspring \leftarrow$ **Multi-directional Mutate**$(c_k, c_{k+\frac{n_p}{2}}, \tilde{f}_{\text{all}})$
17:             Add $offspring$ to $C_i$
18:         **end if**
19:     **end for**
20:     Sample new tasks: $n_t \times \alpha$ from $C_{i+1}$
21:     Sample old tasks: $n_t \times (1 - \alpha)$ from $C_j$, $j = 0$ to $i$
22:     Combine new and old tasks for training
23:     Update $\theta$ using MAPPO
24: **end for**

---

result in uneven development of individual strategies. Consequently, using the previously mentioned encoding method, operating on intermediate tasks at the personal level within the MAS is essential. In a particular round of intermediate task generation, assume that $N$ individuals from the prior task set $T = t_1, t_2, \ldots, t_N$ are randomly split into two groups, $T_A$ and $T_B$. From these groups, we select $N/2$ task pairs $t^A, t^B$ from $T_A$ and $T_B$, respectively, to generate new offspring for the task population.

$$\begin{cases} t_i^{A*} \leftarrow t_i^A + S_i\overrightarrow{D_i}, \\ t_i^{B*} \leftarrow t_i^B + S_i\overrightarrow{D_i} \end{cases}, \quad i = 1, 2, \ldots, \frac{N}{2} \tag{6}$$

In the formula above, $s_i$ denotes the crossover step size for pair $i$, while $\overrightarrow{D_i}$ indicates the crossover direction for pair $i$. The calculations for both $s_i$ and $\overrightarrow{D_i}$ are provided as follows:

$$s_{c,i} = \frac{|\tilde{f}(t_i^A) - \tilde{f}(t_i^B)|}{\max(\tilde{f}(T)) - \min(\tilde{f}(T))} \tag{7}$$

$$\overrightarrow{D_i} = [D_{i,1}, D_{i,2}, \ldots, D_{i,n}]$$

$D_{i,j}$ denotes the direction of the $j$-th agent in pair $i$, obtained by uniform random sampling.

$$D_{i,j} = \begin{cases} 0, & \text{if random variable } \delta_j < 0.5 \\ \theta_{i,j}^A - \theta_{i,j}^B, & \text{if random variable } \delta_j \geq 0.5 \end{cases} \tag{8}$$

## 5 EXPERIMENT

### 5.1 MAIN RESULT

In this section, we assess the effectiveness of CCL on five cooperative tasks in two distinct environments: the simple and complex propagation tasks, along with the Push-ball task from the popular

MPE multi-agent reinforcement learning benchmark (Lowe et al., 2017), as well as the ramp-passing and lock-back tasks from the Hns environment (Baker et al., 2019) based on the MuJoCo framework. Each experiment operates under a 0-1 sparse reward structure, with results validated using three random seeds for each task. The computational setup includes a single Nvidia GeForce RTX 3090 GPU and a 14-core CPU to facilitate the training of deep reinforcement learning models. For policy training, we utilize the MAPPO algorithm (Yu et al., 2022). Moreover, we incorporate recent innovations, such as improving agent decoupling through the integration of attention mechanisms (Vaswani et al., 2017), to further enhance the performance of CCL.

We compared CCL against five baseline methods: **(1). Vanilla MAPPO** (Yu et al., 2022): The training is conducted directly on the target task without any intermediate task generation.
**(2). POET** (Wang et al., 2019): To ensure experimental fairness while preserving the core of the original method, we employ the same coding techniques used in CCL.
**(3). GC** (Song & Schneider, 2022): An enhanced version of POET, introducing innovations for generating intermediate tasks.
**(4). GoalGAN** (Florensa et al., 2018): A baseline enhanced with attention mechanisms for improved performance.
**(5). VACL** (Chen et al., 2021): Utilizes variational methods to generate robust intermediate tasks as baseline comparisons.

We evaluated all the algorithms across several multi-agent environments, assessing the performance of four agents in both simple and challenging scaling scenarios and the cooperation of two agents in a ball-pushing task. Under baseline conditions without curriculum learning, the algorithms struggle to learn effective policies. In contrast, CCL demonstrates superior performance regarding both training speed and outcomes. Most algorithms underperform in the more complex HnS environment, whereas CCL achieves over 95% high performance. The key results are summarized in Table 1 and Table 2, with errors represented as ± standard deviation.

In the MPE environment, we evaluated the number of training steps needed for CCL to achieve optimal performance compared to other baseline approaches. However, in the more challenging HnS environment, specific algorithms were excluded from the comparison because some baseline strategies did not converge, making direct comparisons infeasible.

## 5.2 Ablation Studies

**Adaptive Mutation Step:** As mentioned earlier, an adaptive mutation step size provides more flexibility than a fixed step size, especially for more straightforward expansion tasks. Our ablation studies evaluated three conditions: adaptive mutation step size, fixed mutation step size, and no mutation. While the mutation operation generally increases strategy diversity within the population, an unsuitable mutation step size in sparse reward environments can adversely affect CCL's performance. Notably, with the adaptive mutation step size, the mutation operation's effectiveness in CCL matches that of crossover and variation techniques focusing solely on individual perspectives, as shown in Fig. 3.

**Non-linear Factor in Fitness Function:** As shown in Fig.4, the sigmoid fitness function delivers better performance than the linear form $\tilde{f} = -k|r - 0.5|$. This improvement stems from the sigmoid function's properties: as the agent's success rate approaches 0 or 1, the task's suitability to the agent's abilities decreases exponentially. Specifically, when the success rate is exactly 0.5, the fitness value remains consistently at 0.5. This approach effectively integrates nonlinear elements into the success rate distribution, enabling the fitness function to more accurately represent the relationship between task difficulty and the agent's skill level.

## 6 CONCLUSION

This paper explores the difficulties faced by Multi-Agent Systems (MAS) in sparse reward settings. It introduces a new adaptive curriculum learning algorithm called CCL, developed using a co-evolutionary framework. CCL creates a separate population of intermediate tasks and integrates nonlinear factors into the task evaluation process, ensuring more stable training for MAS. Additionally, the algorithm utilizes an elite prototype fitness evaluation strategy, significantly reducing the computational overhead of assessing population fitness. A key innovation in CCL is the varia-

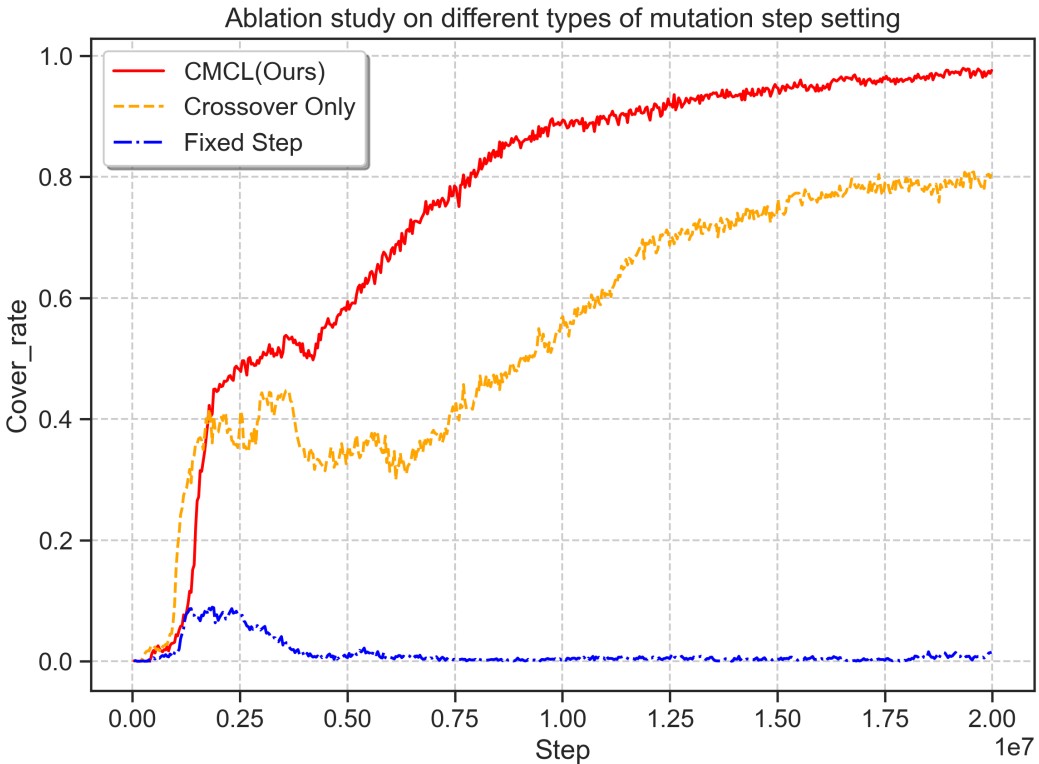

Figure 3: The adaptive step usage ablation experiments which shows its effect.

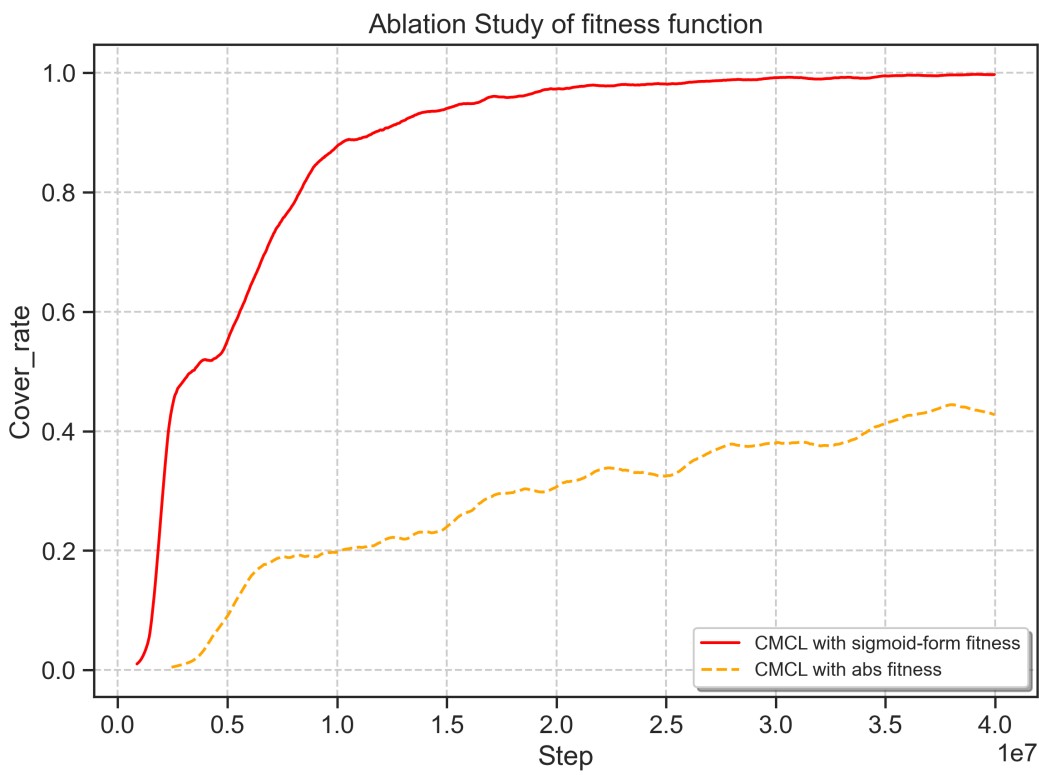

Figure 4: The comparison of using absolute value and sigmoid-shaped fitness function.

Table 1: The Performance Comparison of CCL and Other Baselines on Simulated Environments

| Method | Ramp-Use num_agent = 2 | Lock and Return num_box = 2 num_agent = 2 | Simple-Spread num_agent = 4 num_landmark = 4 | Hard-Spread num_landmark = 4 num_agent = 4 | Push-Ball num_agent = 2 num_ball = 2 num_landmark = 2 |
|---|---|---|---|---|---|
| MAPPOYu et al. (2022) | < 1% | < 1% | < 1% | < 1% | 2% ± 0.5% |
| GCSong & Schneider (2022) | 37.2% ± 18.6% | 8.7% ± 3.2% | 65% ± 12.1% | 79% ± 15.6% | 59% ± 12.3% |
| POETWang et al. (2019) | < 1% | < 1% | 44% ± 9.7% | 10% ± 8.1% | 80% ± 8.4% |
| GoalGANFlorensa et al. (2018) | 9.2% ± 4.2% | < 1% | 82% ± 0.9% | 86% ± 8.8% | 61% ± 8.7% |
| VACLChen et al. (2021) | 94.7% ± 0.8% | 95.4% ± 0.1% | 90% ± 1.6% | 91% ± 6.9% | 90% ± 3.0% |
| **CCL (Ours)** | 98.4% ± 0.3% | 99.1% ± 0.7% | 99% ± 0.2% | 95% ± 3.4% | 96% ± 1.5% |

Table 2: Performance Metrics for Various Methods across Different Tasks

| Method | Simple-Spread | Push-Ball | Hard-Spread |
|---|---|---|---|
| MAPPOYu et al. (2022) | $> 5e7$ | $> 1e8$ | $> 1e8$ |
| GCSong & Schneider (2022) | $> 5e7$ | $> 1e8$ | $1e8$ |
| POETWang et al. (2019) | $> 5e7$ | $> 1e8$ | $1e8$ |
| GoalGAN (att)Florensa et al. (2018) | $> 5e7$ | $1e8$ | $1e8$ |
| VACLChen et al. (2021) | $> 5e7$ | $1e8$ | $1e8$ |
| **CCL (Ours)** | $2e7$ | $6e7$ | $7e7$ |

tional individual view operator, which decouples intermediate task creation from reward reliance in multi-agent cooperation tasks with sparse rewards. CCL demonstrates superior performance across multiple simulation environments, such as MPE and HnS, consistently surpassing existing baseline methods in collaborative tasks. Ablation studies further confirm each component's critical role and effectiveness within CCL.

Although CCL has demonstrated notable advantages in multi-agent cooperation, its application still needs some limitations that future research must address. Current studies have primarily concentrated on multi-agent systems with intensive cooperative behaviors, and further verification and optimization of CCL's performance are required to enhance its adaptability in competitive scenarios or environments with mixed behaviors. Such complex environments introduce challenges like conflicting objectives and the delicate balance between cooperation and competition among agents. Additionally, CCL retains a population of historical courses to support the soft selection mechanism, which improves performance and increases space overhead. Future research should focus on enhancing the storage efficiency of historical data, aiming to mitigate this resource demand through innovative techniques such as data compression, indexing, or selective retention strategies.

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
