# OpenReview forum: "Enhancing Cooperative Problem-Solving in Sparse-Reward Systems via Co-evolutionary Curriculum Learning"
_ICLR.cc/2025/Conference — Submitted to ICLR 2025_

### Official Review · Reviewer_ZBCn · 2024-10-23

**Soundness:** 2
**Presentation:** 2
**Contribution:** 2
**Rating:** 3
**Confidence:** 4

**Summary:**

This paper introduces a novel approach called Collaborative Multi-dimensional Course Learning (CCL) for multi-agent cooperation scenarios to tackle the sparse reward challenge. This approach combines automatic curriculum learning technology, and uses a co-evolutionary framework to develop. The authors claim that CCL outperforms existing baseline methods across multiple simulation environments, such as MPE and HnS.

**Strengths:**

1. The sparse reward MARL studied in the paper is within the scope of ICLR, and relevant to the RL community.
2. The approach appears reasonable and effective.

**Weaknesses:**

1.In general, the novelty and the technical contribution of this work are quite limited. The proposed approach combines various existing approaches, making it challenging to discern its novelty. There is a lack of clarity regarding the connection and differentiation from existing methods. Current techniques such as automatic curriculum learning and evolution algorithm are commonly employed in recent sparse-reward papers. I think the authors should summarize the unique aspects of their approach compared to these existing methods, especially the advantages of combining two technologies.

2.Currently, the experiment section only describes the results without any insights or analysis, which makes the experiment section less helpful. The key results only use Table type to summarize, so why not use more intuitive Figures to display performance? The figures of ablation result only have lines without shaded area, is this right?

3.The writing quality is subpar. The layout of images and tables in the text is chaotic, making it difficult to match the content. There is no corresponding introduction in the main text regarding the detailed content of Fig.1 and Fig.2. In addition, the sub-section titles on L233 and L243 are the same.

**Questions:**

I have asked my questions in the Weaknesses section. I have no further questions.

---

### Official Review · Reviewer_dRFy · 2024-10-28

**Soundness:** 1
**Presentation:** 1
**Contribution:** 1
**Rating:** 3
**Confidence:** 5

**Summary:**

This paper introduces an evolutionary curriculum learning approach aimed at mitigating the sparse-reward issue in multi-agent learning environments. In general, the quality of the paper, including the presentation, the method and the experiments, is pretty low.

**Strengths:**

The research question addressed in this paper—leveraging curriculum learning to enhance reinforcement learning—is both relevant and significant for broader reinforcement learning tasks.

**Weaknesses:**

- The presentation lacks clarity and structure.
-  The use of an evolutionary algorithm in the proposed method lacks novelty.
- The experimental setup is insufficient. It would be more informative to present training plots instead of tables to illustrate performance. The inclusion of timesteps in Table 2 is unconventional and raises questions about clarity. If baseline methods without curriculum learning also achieve optimal results, this suggests that sparse rewards may not be a significant issue for the given tasks.

**Questions:**

See the Weaknesses.

---

### Official Review · Reviewer_nTYk · 2024-10-28

**Soundness:** 3
**Presentation:** 2
**Contribution:** 1
**Rating:** 3
**Confidence:** 3

**Summary:**

This paper attempts to address the issue of sparse-reward problem which can make a multi-agent learning problem much more difficult. Using a co-evolutionary algorithm, the authors attempt to generate intermediate tasks to each agents as well as co-evolve the environment and the agents to enhance training in sparse-reward multi-agent environment. The authors include an empirical comparison with the several curricular RL algorithms in multi-agent benchmarks such as the Multi-agent Particle Environment (MPE).

**Strengths:**

The authors clearly mark why sparse reward problem can especially be a challenge in multiagent environment.

**Weaknesses:**

- While the authors have delivered the motivations of the paper quite well, I feel that the paper's drawback is the lack of theoretical foundations. There is a lack of theoretical background on why the author's algorithm would work well. In addition, I would like to hear more about why the authors decided to choose a specific type of structure when designing their algorithm. What are the motivations of the decisions? any engineering or theoretical considerations?

- While the authors have included several notable baselines in the recent years, I think it would have been nicer to include more baselines in the multiagent domain as this paper is about multiagent domain. There are several useful works, such as Policy Space Response Oracle (Lanctot et al, 2017) if the authors are looking for baselines with more theoretical background, as well as papers in multi-robot teaming if the authors are looking for baselines with more engineering based decisions.

- I think the overall writting can be improved as well. While there weren't that many major typos or incomplete sentences that made reading difficult, I think the structure can be improved. For example, in line 246, the authors mention that the details of MAPPO will not be discussed in this paper, and says the same thing again in less then 10 lines down.

**Questions:**

- I think my opinion would change the greatest if the authors can provide a theoretical background behind the design choices they have made and/or if they can compare their ideas with curricular approaches in multiagent domain.

- How is crossover done in this paper? How are tasks represented and encoded in this paper?

- In line 249, what is 'better initial policy performance' referring to? Better than what? and why does it has to be better?

- Figure 3 is not showing standard deviations between the graph.

- In regards to using evolutionary algorithms for curriculum design in multiagent environments, I wonder how the author's paper would compare to this relatively recent paper, Genetic Algorithm for Curriculum Design in Multi-Agent Reinforcement Learning by Song et al, 2024. The paper uses evolutionary algorithms for curriculum generation in multiagent environments and has been shown to work quite well in collaborative setups with the MPE benchmarks so I think it would be a interesting to see how the submission 13141 compares with this one. Whether the authors of submission 13141 are going for a revision or a resubmit, this paper would be an interesting related work. Please note that this paper is quite recent and while incorporating this paper in to the revision will certainly improve my score, I haven't and will not reduce my review score for not including this paper as the paper was accepted in CoRL only a month before the ICLR deadline.

---

### Official Review · Reviewer_HxZk · 2024-11-12

**Soundness:** 1
**Presentation:** 1
**Contribution:** 2
**Rating:** 5
**Confidence:** 5

**Summary:**

This paper primarily integrates co-evolution and curriculum learning to address the issues brought by the quality of reward signals in MARL and the increased number of agents. The proposed CMCL method evolves the curriculum to solve the target tasks more efficiently. CMCL outperforms the baselines in MPE and Hide and Seek.

**Strengths:**

- Integrating curriculum learning with evolutionary algorithms to solve the difficult problem of multi-agent cooperation in MAS is a promising research direction.

**Weaknesses:**

- The logical structure of the paper needs further improvement. It is recommended that the author provide a clear chapter arrangement and introduction at the beginning of Chapter 4.
- The experimental section is relatively weak, the environment tasks are too simple, and no learning curves are provided. Figures 3 and 4 do not provide the variance.
- The overall writing of the paper has many issues. Many parts need thorough proofreading to make the paper more rigorous. For example, it should be f^{\widetilde}_{all}. The font bolding within formulas and texts is inconsistent, formula 8 lacks proper spacing, and the reasoning behind why this design is effective has not been clearly explained.
- The ablation study lacks results for using only mutation. CMCL has many hyperparameters that need to be tuned, but the paper does not provide an analysis.

**Questions:**

- In Formula 3, why can we directly measure the agents and tasks? Does this mean the distance between states? This seems only feasible in some simple tasks.
- In Formula 4, Why is it designed this way?
- Can the author provide learning curves? It seems that only the ablation study provides curves.
- Why are Variational Individual-perspective Crossover and Mutation more efficient?
- Other works that combine evolutionary algorithms and MARL should be introduced and compared in the related work section [1].

If the author can address my questions, I will consider raising the score, but the current version does not meet NeurIPS's high standards.

---

[1] RACE: Improve Multi-Agent Reinforcement Learning with Representation Asymmetry and Collaborative Evolution

---

### Meta-Review · Area_Chair_WwRt · 2024-12-19

**Metareview:**

The reviewers generally agree that the paper's exposition is unclear, and the proposed evolutionary approach combines multiple existing methods without providing any ablations to understand which of these components contributes to performance of the method. There is thus much room for improvement in this work, making it unfit for publication at this time.

**Additional Comments On Reviewer Discussion:**

Reviewers largely agreed in that the paper does not provide sufficient information about how their the various components of the proposed evolutionary method contribute to its performance. In the absence of such details, it is unclear what is the novel contribution of this work, as the constituent parts of the proposed method are themselves not novel.

---

### Decision · Program_Chairs · 2025-01-22

Reject